# When Fair Ranking Meets Uncertain Inference

Avijit Ghosh
Northeastern University
avijit@ccs.neu.edu

Ritam Dutt
Carnegie Mellon University
rdutt@andrew.cmu.edu

Christo Wilson
Northeastern University
cbw@ccs.neu.edu

## ABSTRACT

Existing fair ranking systems, especially those designed to be demographically fair, assume that accurate demographic information about individuals is available to the ranking algorithm. In practice, however, this assumption may not hold — in real-world contexts like ranking job applicants or credit seekers, social and legal barriers may prevent algorithm operators from collecting peoples' demographic information. In these cases, algorithm operators may attempt to infer peoples' demographics and then supply these inferences as inputs to the ranking algorithm.

In this study, we investigate how uncertainty and errors in demographic inference impact the fairness offered by fair ranking algorithms. Using simulations and three case studies with real datasets, we show how demographic inferences drawn from real systems can lead to unfair rankings. Our results suggest that developers should not use inferred demographic data as input to fair ranking algorithms, unless the inferences are extremely accurate.

## CCS CONCEPTS

• **Social and professional topics → Codes of ethics**; • **Information systems → Retrieval models and ranking**.

## KEYWORDS

ranking algorithms, demographic inference, algorithmic fairness, ethical ai, noisy protected attributes, uncertainty

**ACM Reference Format:**
Avijit Ghosh, Ritam Dutt, and Christo Wilson. 2021. When Fair Ranking Meets Uncertain Inference. In *Proceedings of the 44th International ACM SIGIR Conference on Research and Development in Information Retrieval (SIGIR '21), July 11–15, 2021, Virtual Event, Canada.* ACM, New York, NY, USA, 11 pages. https://doi.org/10.1145/3404835.3462850

## 1 INTRODUCTION

Social biases in algorithms have been investigated and identified in a number of contexts [20, 21, 33, 39, 45, 46, 49, 53, 62, 66–68]. As a result, there is now a thriving research community that seeks to develop fair algorithms [7, 15, 34, 40, 80] and efforts by activists and regulators to see that these tools are adopted in practice [24].

In cases where people are the data subjects being input to classification or ranking algorithms, the vast majority of existing work assumes that ground-truth demographic information will be available to mitigate sexism, racism, ageism, and other social biases [23]. This demographic data is crucial as it is used to measure and control for unfair biases, thus enabling fair outcomes.

Unfortunately, this assumption about the availability of ground-truth demographic data is often violated in practice. For example, in real-world contexts like assessing job applicants or credit seekers, social and legal barriers may prevent algorithm operators from collecting peoples' demographic information [3, 10].

The unavailability of ground-truth demographic data has led some system developers to adopt an alternative approach: infer protected class information from data and then supply it to the fair algorithm as input. One example of this is the Bayesian Improved Surname Geocoding (BISG) inference algorithm that is used by lenders and health insurers in the U.S. to infer people's race and ethnicity [1, 14]. This demographic data is used to ensure that lenders are making race-neutral lending decisions and that health insurers are not discriminating based on race. Given the high-stakes of these use cases, it is clear that accurate demographic information is critical, lest unchecked discrimination lead to serious harms.

The use of inferred data raises the issue that errors in inference may subvert the fairness objectives that a fair algorithm is attempting to optimize for. Intuitively, a fair algorithm cannot be expected to control for social biases if those biases are not represented in data due to errors. To the best of our knowledge, this problem has not been explored systematically in the literature, despite the fact that consequential real-world systems like BISG have already adopted the practice.

In this study we investigate how uncertainty in demographic inference impact fairness guarantees in the context of ranking algorithms. We approach this question using two complementary techniques. *First*, we use simulations to explore the relationship between population demographics, fairness metrics, and inference errors under controlled conditions. *Second*, to address the issue of ecological validity, we examine three case studies based on real-world datasets. Each of these datasets includes ground-truth demographic data, which enables us to generate a baseline unfair ranking and an "optimal" fair ranking. We compare these lower and upper bounds against rankings generated by a fair ranking algorithm when using erroneous demographic inferences as input. We present results using demographic inference error rates drawn from five real-world algorithms.

Our results suggest that developers should not use inferred demographic data as input to fair ranking algorithms, unless the inferences are extremely accurate. Our simulations and case studies make clear that errors in inference can dramatically undermine fair ranking algorithms, causing them to produce rankings that are much closer to the unfair baseline than the optimal fair ranking. We even observe instances where erroneous inferences cause groups that were not disadvantaged in the baseline unfair ranking to become disadvantaged after a "fair" re-ranking.

## 2 RELATED WORK

We begin by briefly discussing fairness issues with machine learning algorithms in general, and ranking algorithms in particular. We also touch on documented problems with demographic-based classification and inference.

**Algorithmic Fairness.** Hype around big data and "artificial intelligence" has sharpened concerns about the social impact of these systems. Barocas and Selbst [4] and Osoba and Welser IV [63] discuss the ways that socio-technical systems may perpetuate unfair biases in various contexts. Researchers in academia and industry are building tools to help detect biases in algorithms [6, 52, 65, 77]. Additionally, there is a growing library of fairness-aware machine learning algorithms for classification [31, 38, 43, 57], regression [2, 7], causal inference [51, 60], word embeddings [11, 12], machine translation [25] and finally, ranking [18, 73, 81].

**Fair Ranking.** Several fair ranking algorithms have been proposed in the literature. Early approaches only attempted to make a ranked list fair between two groups [18, 81]. More recent methods use constrained learning and treat fair ranking as an optimization problem [73]. Other approaches achieve fairness through pairwise comparisons [8] or apply fairness constraints in learning-to-rank methods [58, 82].

**Fair Ranking Metrics.** Mehrabi et al. [55] provide an extensive list of fairness definitions that appear in the fair machine learning literature. This includes concepts like *equalized odds* and *equal opportunity* [34], *demographic parity*, *treatment parity* [23], etc.

These concepts have been adapted specifically to the domain of ranking. Metrics developed by Yang and Stoyanovich [78] measured the underlying population representation in the top-ranked items, while other metrics such as those by Singh and Joachims [73] and Sapiezynski et al. [71] conceptualized ranking fairness as an attention or exposure allocation problem to the different subgroups. Singh and Joachims [73] propose that attention allocation metrics correspond to the problem of *disparate impact* since top-ranked results gain more attention than results at the bottom.

Another consideration is the cardinality of the protected categories. Several metrics from earlier literature, such as those by Zehlike et al. [81] and Kuhlman et al. [48], are binary, i.e., they are only able to assess fairness between two groups. Binary metrics cannot be used if intersectional fairness is desired, e.g., fairness between White males and White females. Newer metrics like those proposed by Geyik et al. [29] that compare entire population distributions over an unspecified number of subgroups, or attention-based metrics [9, 71, 73] that also deal with the population distributions, are agnostic to group cardinality, and thus lend themselves to intersectionally fair frameworks [26, 30].

Problematically, a meta-analysis of these (and other) fair ranking metrics has shown that they often disagree about whether a given ranked list is "fair" [64]. This motivated us to choose several metrics for our study, as we describe in § 3.2.2.

**Inferred Attributes.** There are examples in the literature that highlight accuracy problems with demographic inference algorithms, perhaps most notably when Buolamwini and Gebru [13] showed how the accuracy of facial analysis systems at predicting gender fell when presented with images of darker-skinned people.

Geiger et al. [28] note that leveraging crowd workers to produce demographic data is also problematic, and work on the best ways to collect such data is only beginning to emerge [42].

The interactions between noise in protected attribute data and algorithms trying to ensure fairness is sparsely studied despite its potentially far-reaching consequences. There have been studies on the stability of classification algorithms with noisy data [69]. Friedler et al. [27] note that classifiers may not be stable in the face of variations in the training dataset. Chen et al. [19] analyzed disparity under unobserved protected attributes using demographic inference, but they do not study the impact of inference on the fairness providing algorithm itself.

Recent studies have proposed frameworks to achieve fair classification [16] and fair subset selection [56] despite noise in inferred protected attributes. However, to the best of our knowledge, no work has looked at how noisy or imperfectly inferred protected attributes impact fair ranking tasks.

## 3 ALGORITHMS AND METRICS

In this study, our goal is to assess how well fair ranking algorithms are able to achieve their stated fairness objectives when given input data that includes ground-truth and inferred demographic information. Rather than approaching this question theoretically, we do so empirically using simulations and case studies. To implement these empirical experiments we require: (1) one or more fair ranking algorithms to evaluate, (2) fair ranking metrics that encompass a spectrum of fairness definitions, and (3) error rates drawn from inference algorithms. To improve confidence in our results, we strive to evaluate algorithms and datasets that are drawn from real-world deployments.

With these goals and guiding principles in mind, we now move on to selecting algorithms and metrics.

### 3.1 Fair Ranking Algorithm

The fair ranking algorithm we chose for this study was developed by Geyik et al. [29] from LinkedIn. Their paper presents four different re-ranking algorithms with varying stability but with one central goal: to achieve the desired distribution of population in the top-ranked results with respect to one or more protected attributes. At a high-level, the algorithm takes an unfairly arranged list and an integer $K$ then generates a fairness-aware list of the top $K$ candidates such that the fraction of candidates in each subgroup matches their fraction in the underlying population. While other algorithms from prior work [18, 73, 81] have similar goals, this algorithm was extensively tested and deployed in LinkedIn's Talent Search system. The authors of the paper claim that the deployment led to *"tremendous improvement in the fairness metrics (nearly three-fold increase in the number of search queries with representative results) without affecting the business metrics, which paved the way for deployment to 100% of LinkedIn Recruiter users worldwide"* [29]. Since our work focuses on the possible breakdown of fair ranking algorithms in real-world, deployed scenarios, this work was the best fit for our research purposes.

Of the four algorithms presented in the Geyik et al. [29] paper, we chose the Deterministic Constrained Sorting algorithm or *DetConstSort* as our benchmark fairness algorithm since it is

theoretically proven to be feasible for protected attributes having a large number of possible attribute values, unlike the other three greedy fair ranking algorithms in the paper.

*DetConstSort* creates a ranked list of candidates, such that for any particular rank $k$ and for any group attributes $g_j$, the attribute occurs at least $\lfloor p_{g_j}.k \rfloor$ times in the ranked list ($p_{g_j}$ = proportion of members in the list belonging to $g_j$). However, unlike other fair ranking algorithms that greedily pick the best candidate for a particular rank, the *DetConstSort* algorithm also strives to improve the sorting quality by re-ranking the candidates that come above it (so that candidates with better scores are placed higher in the list), as long as the resultant list satisfies the feasibility criteria. Thus, the algorithm can be conceptualized as solving a more general interval constrained sorting problem. Since the *DetConstSort* algorithm is constrained to be feasible it optimizes the Skew and NDKL fairness metrics, which we introduce in the next section.

## 3.2 Metrics for Ranking Evaluation

The second decision we needed to make to accomplish our study was choosing metrics for evaluating the fairness of representation in ranked lists. We focus on metrics that (1) assess *group fairness* [23], possibly balanced against secondary objectives, and (2) are capable of dealing with multiple subgroups (i.e., not just binary protected versus unprotected classes). For our analysis, we adopted the definition of a subgroup as a Cartesian product of $\geq$ 2 groups, as defined in Ghosh et al. [30]. A subgroup $sg_{a_1....a_n}$ is defined as set containing the intersection of all members who belong to groups $g_{a_1}$ through $g_{a_n}$, where $a_1, a_2...a_n$ are marginal protected attributes like race, gender, etc. Notation wise:

$$sg_{a_1 \times a_2 \times ... \times a_n} = g_{a_1} \cap g_{a_2}... \cap g_{a_n}. \tag{1}$$

Note that if the metrics satisfy fairness for a set of subgroups they will also be fair for the constituent marginal groups [26].

### 3.2.1 Representation-based Metrics.
To get an overall sense of group fairness in a given ranked list, we chose two (slightly modified) representation-based metrics introduced by Geyik et al. [29]. These metrics do not incorporate attention, i.e., they assess the representation of people from different groups based solely on how many of those people appear in the list relative to the underlying population. The first metric is computed per group, while the second is aggregated across groups.

**Skew.** Given a ranked list $\tau$, the Skew for attribute value $sg_i$ at position $k$ is defined as

$$\text{Skew}_{sg_i}@k(\tau) = \frac{p_{\tau^k,sg_i}}{p_{q,sg_i}} \tag{2}$$

where $p_{\tau^k,sg_i}$ represents the proportion of members belonging to subgroup $sg_i$ within the top $k$ items in the ranked list $\tau$, and $p_{q,sg_i}$ represents the proportion of members belonging to subgroup $sg_i$ in the overall population $q$. Ideally, $\text{Skew}_{sg_i}@k$ should be close to one for each $sg_i$ and $k$, indicating that people from $sg_i$ are represented in $\tau$ proportionally relative to the underlying population. $Skew_{sg_i}@k > 1$ denotes that the subgroup $sg_i$ is over-represented among the top $k$ candidates, and vice versa when the $Skew_{sg_i}@k < 1$.

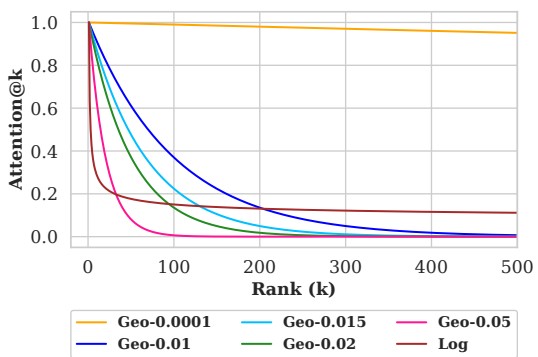

**Figure 1: Attention versus rank for six attention functions.**

**Divergence.** Given a ranked list $\tau$, the Normalized Discounted Kullback–Leibler (NDKL) Divergence is defined as

$$\text{NDKL}(\tau) = \frac{1}{Z} \sum_{i=1}^{|\tau|} \frac{1}{log_2(i+1)} d_{KL}(D_{\tau^i}||D_r) \tag{3}$$

where $d_{\text{KL}}(D_1||D_2) = \sum_j D_1(j) log_2 \frac{D_1(j)}{D_2(j)}$ is the KL divergence score of distribution $D_1$ with respect to distribution $D_2$ and $Z = \sum_{i=1}^{|\tau|} \frac{1}{log_2(i+1)}$. NDKL can be interpreted as a weighted average of the logarithm of the Skew scores for all the groups in a ranked list. NDKL values close to zero indicate that people from all subgroups are represented proportionally in a given ranked list, since the KL-Divergence of the population between the top $k$ candidates and the underlying population will be zero. A large difference in the distributions of the different groups in the top $k$ ranked candidates leads to a higher NDKL score.

### 3.2.2 Attention-based Metrics.
Studies have repeatedly shown that people do not pay equal attention to all items in ranked lists [59, 61]; rather, peoples' attention decays as they progress down the list, eventually abandoning the task entirely. This observation suggests that using overall representation to assess fairness is misleading, since (1) people may not look at all available items and (2) they pay more attention and are thus more likely to act on higher ranking items. To take attention into account, we computed attention per group and in aggregate across groups like in § 3.2.1.

**Attention.** In this study, we adopted the geometric distribution to model decay in attention, similar to prior work by Sapiezynski et al. [71]. We compute attention at $k$ as

$$\text{Attention}_p@k(\tau) = 100 \times (1-p)^{k-1} \times (p) \tag{4}$$

where $\tau$ is the ranked list and $p$ represents the proportion of attention provided to the first result. For our experiments we set $p = 0.015$ because at this value attention decays to zero at $k = 300$, which is the value of $k$ we fix for our experiments. Although most prior work in the Information Retrieval (IR) literature uses logarithmic decay to model attention [73, 78], we did not adopt it because it models attention decay at an unrealistically slow rate [68] and its shape flattens out at low ranks. Figure 1 shows how attention decays as a function of rank for a variety of values of $p$, as well as under logarithmic decay.

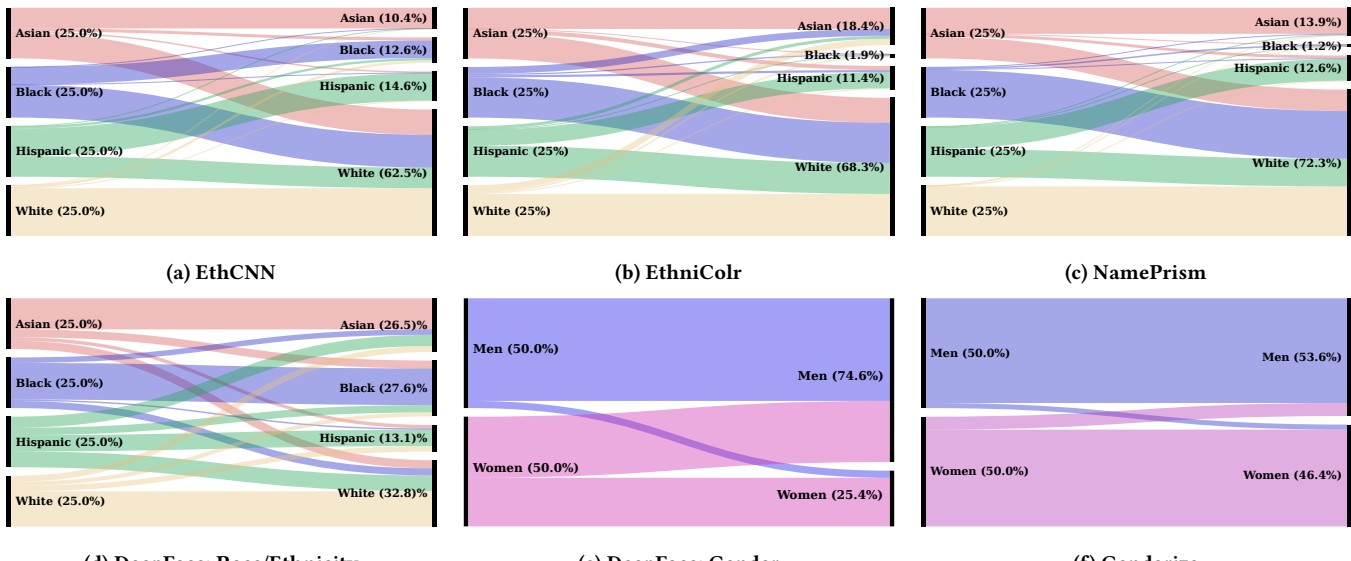

**Figure 2: Sankey plots showing the distribution of ground-truth (left) and inferred (right) demographic traits for five algorithms. The algorithms tend to mis-classify minorities as Whites. DeepFace tends to mis-classify Women as Men.**

The $i^{th}$ element in $\tau$ has an associated score, denoted $s_i^\tau$, that corresponds to the utility or relevance of the item, and a subgroup-attribute value, denoted by $sg_i^\tau$. The elements in the ranked list are arranged in decreasing order of score such that $s_i^\tau \geq s_j^\tau \ \forall \ i \leq j$. We define

$$\eta_{sg_j,\tau} = \frac{1}{|sg_j|} \sum_{i=1}^{|\tau|} \text{Attention}_p@i \ \forall \ sg_i^\tau \in sg_j \quad (5)$$

where $\eta_{sg_j,\tau}$ denotes the mean attention score of the $sg_j$ protected attribute for $\tau$ and

$$\text{ABR}_\tau = \frac{\min_{sg_j}(\eta_{sg_j,\tau})}{\max_{sg_j}(\eta_{sg_j,\tau})} \quad (6)$$

where $\text{ABR}_\tau$ or the Attention Bias Ratio for the ranking $\tau$ quantifies the disparity between the groups with the lowest and highest mean attention score ($\eta_{sg_j,\tau}$). $\text{ABR}_\tau = 1$ is the ideal score, i.e., all subgroups (and thereby all groups) receive equal attention.

**3.2.3 Ranking Quality Metrics.** Classic IR literature has proposed several evaluation metrics to measure the ranking quality of an IR system [54]. We measure two different metrics in our study: a cumulative gain based metric, and a rank change metric to measure loss in ranking utility.

**Normalized Discounted Cumulative Gain.** NDCG is a widely used measure to evaluate search rankings [41].

$$\text{NDCG}(\tau) = \frac{1}{Z} \sum_{i=1}^{|\tau|} \frac{s_i^\tau}{log_2(i+1)} \quad (7)$$

where $s_i^\tau$ is the utility score of the $i^{th}$ element in the ranked list $\tau$ and $Z = \sum_{i=1}^{|\tau|} \frac{1}{log_2(i+1)}$.

**Rank Change.** This metric is a measure of the amount of itemwise distortion from the original list to the fairness-aware re-ranked list, much like the ranking utility loss measure in Zehlike et al. [81]. We define Rank Boost as the boost in the rank of an item due to re-ranking. For a candidate $C_A$,

$$\text{Rank Boost}_{C_A} = \tau_{org}[C_A] - \tau_{new}[C_A] \quad (8)$$

where $\tau_{org}$ and $\tau_{new}$ denotes the original and re-ranked list respectively. A positive value indicates that a candidate was assigned a higher rank after re-ranking.

For a subgroup $sg_j$, the Average Rank Change (ARC) is defined as the average of the absolute rank boosts over all candidates in that subgroup:

$$\text{ARC}_{sg_j,\tau} = \frac{1}{|sg_j|} \sum_{i=1}^{|\tau|} |\text{Rank Boost}_{C_i}|; \ \text{where} \ C_i \in sg_j. \quad (9)$$

Finally, we define Maximum Absolute Rank Change (MARC) for a particular list as the maximum value of the ARC over all subgroups in that list:

$$\text{MARC}_\tau = max(ARC_{sg_i,\tau}); \ \forall sg_i \in |sg|. \quad (10)$$

## 3.3 Demographic Inference Algorithms

The final decision we needed to make for this study was selecting demographic inference algorithms. Our intent is to compare the fair rankings generated by the *DetConstSort* algorithm when given ground-truth and inferred demographic information, using the metrics introduced in § 3.2, so as to quantify the impact (if any) of mis-classifications.

We chose five diverse inference algorithms that rely on different features and machine learning techniques. For each algorithm, we computed its confusion matrix when predicting peoples' ethnicity/race and gender (in one case) using ground-truth data

with known demographics. We evaluated the four algorithms in
§ 3.3.1 using voter records from the state of North Carolina,[1] which
are publicly available records that have the name, address, race,
gender, and other personal information of each registered voter in
the state. We evaluated the facial analysis algorithm in § 3.3.2 using
the FairFace dataset [44].

Using these five algorithms we predicted race/ethnicity (Asian,
Black, Hispanic, and White) and gender (man and woman). We
fully acknowledge that these categories are problematic, however,
we adopted them because they are the categories supported by the
inference algorithms from prior work. We discuss the problems and
limitations that derive from these categories in § 6.2.

Figure 2 shows the results of demographic inference using
these five algorithms. We used these confusion matrices in our
experiments to intentionally mis-classify data, so as to observe the
effect on fair ranking performance.

**3.3.1 Name-based Inference.** We chose three algorithms that
attempt to predict peoples' race/ethnicity based on their name, and
we choose one algorithm that does so for gender prediction.

**EthCNN.** We employed a Convolutional Neural Network (CNN)
architecture similar to Kim [47] to infer peoples' ethnicity from their
names, where the name is represented as a sequence of characters.

**EthniColr.** Inspired by the work of Hofstra et al. [36], we
used Ethnicolr,[2] the publicly available library from Sood and
Laohaprapanon [74], to predict an individual's race/ethnicity from
their full name. Ethnicolr employs a neural architecture to model the
relationship between the characters in a name and race/ethnicity.

**NamePrism.** We used the NamePrism API[3] by Ye et al. [79]
for race/ethnicity classification. Motivated by the observation that
individuals frequently communicated with peers of similar age,
language, and location [50], Nameprism exploits the homophily
phenomena in email contact lists to create name embeddings that
can be used to predict race/ethnicity.

**Genderize.** To infer binary gender from names we used a service
called genderize.[4] As of 2021, the dataset underlying genderize
consists of 114,541,298 names collected from 242 countries and
territories. While the sources of the names are not revealed [70],
the site claims that the API has been used for data analysis in articles
from the Guardian, the Washington Post, and other outlets.

**3.3.2 Facial Analysis-based Inference.** We selected one
algorithm that infer demographics from images of faces.

**DeepFace.** We used the public wrapper [72] for DeepFace by
Facebook [76] to obtain DeepFace's error rates when classifying
race/ethnicity and gender from the FairFace dataset [44].

# 4 EXPERIMENTS

In this section we outline the experiments that we performed to
examine the relationship between inferred demographics and fair
ranking.

[1]https://www.ncsbe.gov/results-data/voter-registration-data
[2]https://github.com/appeler/ethnicolr
[3]http://www.name-prism.com/
[4]https://genderize.io/

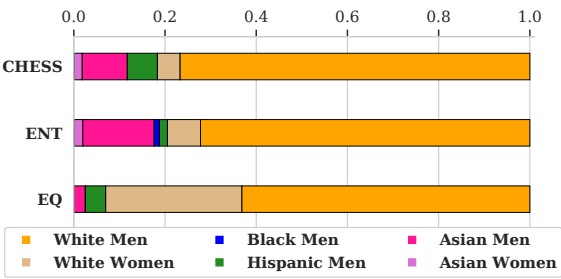

**Figure 3: Population subgroups in our Chess players,
Entrepreneurs, and Equestrians datasets.**

| Distribution | NDKL | ABR |
|---|---|---|
| *Dist A* (W: 0.33, B: 0.33, A: 0.33) | 0.08 | 0.66 |
| *Dist B* (W: 0.2, B: 0.3, A: 0.5) | 0.08 | 0.71 |
| *Dist C* (W: 0.1, B: 0.3, A: 0.6) | 0.30 | 0.86 |
| *Dist D* (W: 0.1, B: 0.2, A: 0.7) | 0.37 | 0.91 |
| *Dist E* (W: 0.25, B: 0.25, A: 0.25, H: 0.25) | 0.11 | 0.60 |
| *Dist F* (W: 0.1, B: 0.2, A: 0.6, H: 0.1) | 0.42 | 0.70 |

**Table 1: Fairness metrics computed between the target
distribution on the left (*A*sian, *B*lack, *H*ispanic, and *W*hite)
and randomly generated unfair distributions. NDCG and
MARC for the unfair lists are 1.0 and 0 in all cases.**

## 4.1 Simulations

In our first experiment, we examined the relationship between
demographic mis-classification and fair ranking guarantees under
controlled conditions by performing simulations using synthetic
data. We used a modified version of the synthetic ranked list
generation method discussed in Geyik et al. [29], as follows:

(1) We manually crafted six ground-truth probability
distributions $P$ for the protected attributes of the simulated
people. The distributions, labeled A through F and shown in
Table 1, each contained three or four groups. These were
the target distributions of our fairly re-ranked lists.

(2) For each probability distribution $P$, we generated 1,000
people per group $g_i \in P$ and assigned each a random utility
score $s_i \in [0, 1]$. We then sorted the combined list of people
in decreasing order of $s_i$ to generate the ranking $\tau$.

(3) We ran the *DetConstSort* algorithm discussed in § 3.1 with
the desired distribution $P$ and $\tau$ as inputs to produce the
fairness-aware re-ranked list $\tau_f$. $|\tau_f| = 300$.

(4) We calculated NDKL, ABR, NDCG, and MARC on $\tau$ and $\tau_f$.

(5) We repeated steps 2–4 100 times and computed the mean
values for our metrics.

(6) We repeated steps 2–5 for demographic prediction accuracies
varying from 0.1 to 1.0. For instance, an accuracy of 0.1 meant
that the attribute $g_i$ was predicted correctly 10% of the time
and therefore, in our simulation, we mis-classify $g_i$ as any
$g_j$ where $j \neq i$ 10% of the time.

Table 1 shows the mean fairness metrics for our empirical
distributions before running *DetConstSort*.

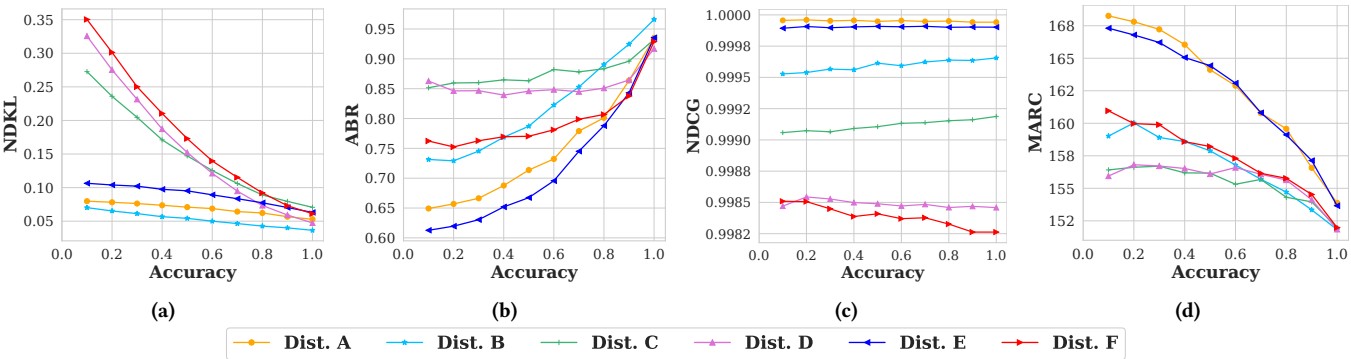

**Figure 4: Distributions of NDKL, ABR, NDCG and MARC scores for fairly-ranked lists as demographic inference accuracy was varied, based on simulations using synthetic data. For details about the ground-truth population distributions, refer to Table 1.**

## 4.2 Case studies

To establish the ecological validity of our study, we perform detailed case studies on three ranked lists obtained from the real world, to measure the potential harms of algorithmic demographic inference on fairness-aware re-ranking tasks.

***4.2.1 Datasets.*** For our case studies, we required datasets that had both names and images for our inference algorithms discussed in § 3.3. We collected these datasets from three publicly available sources on the internet, as discussed below. All three datasets were downloaded in January 2021.[5]

**Chess Rankings.** We downloaded a ranked list of chess players sorted by their World Chess Federation (French: Fédération Internationale des Échecs or FIDE) ratings from the official FIDE website.[6] Along with the ratings, the website also provided the full name, image, and self-identified binary gender of the players.

**Crunchbase Entrepreneurs Ranking.** We downloaded a list from Crunchbase[7] of US startup founders who received Series A funding in the last 5 years. We summed the Series A funds of founders who were part of multiple Series A funding rounds during this time frame. We collected the name, image, and self-identified binary gender of the founders.

**Equestrian Rankings.** We downloaded a ranked list of equestrian athletes arranged by their Fédération Equestre Internationale (FEI) ratings from the official FEI website.[8] Along with the ratings, we also collected the full name, image, and self-identified binary gender of the athletes.

***4.2.2 Data Annotation and Cleaning.*** The datasets we collected in § 4.2.1 contain ground-truth gender information[9] for each individual, but not race/ethnicity. To obtain race/ethnicity information, we followed a similar process as the annotation method for the occupations dataset in Celis and Keswani [17]. We asked workers from Amazon Mechanical Turk to label the images

---

[5]The code and datasets used in this paper can be found at https://github.com/evijit/SIGIR_FairRanking_UncertainInference.

[6]https://ratings.fide.com/

[7]https://crunchbase.com/

[8]https://www.fei.org/jumping/rankings

[9]We consider these gender labels to be ground-truth because they are self-identified. Unfortunately, these organizations force individuals to identify with a binary gender.

| Algorithm | Inference Type | Race | Gender | Fair |
|---|---|---|---|---|
| BASE (Baseline) | None | Perceived | Ground Truth | No |
| ORCL (Oracle) | None | Perceived | Ground Truth | Yes |
| CNNG (EthCnn_Gen) | Name | EthCNN | Genderize | Yes |
| ECLG (Ethnicolr_Gen) | Name | Ethnicolr | Genderize | Yes |
| NPMG (Nameprism_Gen) | Name | Nameprism | Genderize | Yes |
| DPFC (Deepface) | Face image | DeepFace | DeepFace | Yes |

**Table 2: The algorithms and sources of demographic data (ground-truth, perceived, inferred) used in our case studies.**

of faces that we collected. For each image, we asked workers to choose from among the following races/ethnicities based on their best judgment: White/Caucasian (Non Hispanic), Hispanic/Latino, Black/African, Asian (Far East, Southeast Asia, and the Indian subcontinent), or Other/Not sure. Each image was labeled by three independent workers and we accepted the label with majority support. After labeling we dropped 6%, 4%, and 2% of people from our lists, respectively, because they lacked majority consensus. We restricted our task to workers with ≥ 90% approval ratings and our task paid roughly $12/hour.

We do not refer to the race/ethnicity labels that we obtained from crowd sourcing as "ground-truth" because there are no phenotypical determinants of race or ethnicity. Instead, we refer to these labels as "perceived" because they correspond to the perceptions of race and ethnicity held by our workers, as informed and filtered through their own cultural lenses.

For each ranked list, we cleaned the dataset by removing all entries that did not have a picture, and then by removing subgroups that had a population less than 1% of the total length of the list. The final datasets consisted of 3,251 chess players, 3,308 startup founders, and 1,115 equestrian athletes, respectively. Figure 3 shows the population summary statistics for our three datasets, broken down into intersectional subgroups.

***4.2.3 Measurement Approach.*** To analyze the impact of demographic inference on fairness guarantees in our case studies, we used the following approach. First, we computed NDKL, ABR, NDCG and MARC on the original ranked lists that we crawled and the fair re-rankings produced by *DetConstSort* given ground-truth gender and perceived race/ethnicity data. We refer to these as "Baseline" and "Oracle," respectively, with the latter serving as our best-case fairness benchmark. Next, we reran *DetConstSort*

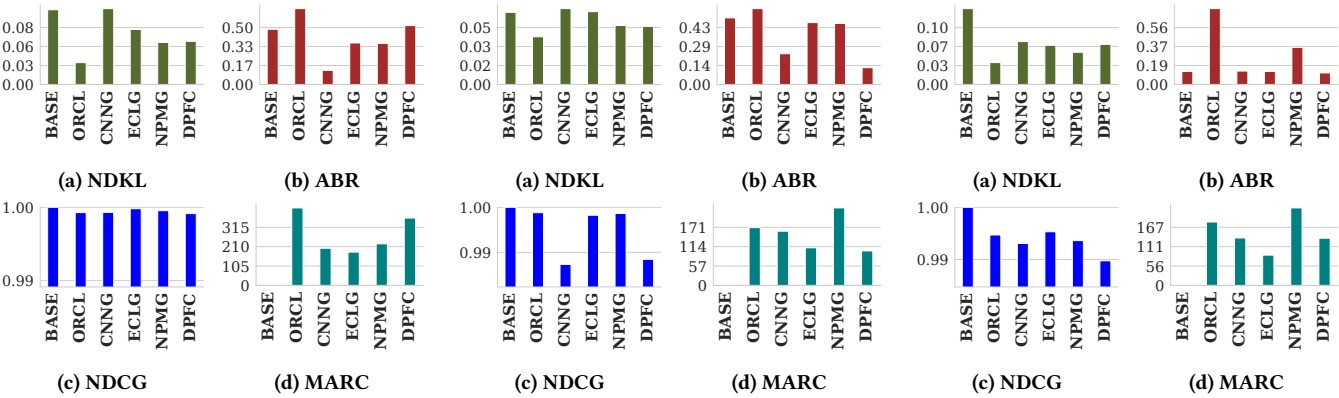

Figure 5: Chess Dataset.    Figure 6: Entrepreneurs Dataset.    Figure 7: Equestrians Dataset.

after introducing demographic mis-classifications (while using the ground-truth gender and perceived race/ethnicity data to compute the fairness metrics NDKL and ABR). The breakdown of the inference algorithms used to produced these re-ranked lists are described in Table 2.

## 5 RESULTS

Having discussed our methods and the structure of our experiments, we now present our results.

### 5.1 Simulations

We present the results of our simulations in Figure 4, from which we make five observations. *First*, both of the fairness metrics suffer in proportion to the error rate of demographic inference. As shown in Figure 4, NDKL falls (i.e., approaches representational fairness), and ABR rises (i.e., approaches attention parity) as the accuracy of demographic inference increases. This result is intuitive: we cannot expect *DetConstSort* to perform at its best when the underlying demographic data is inaccurate.

*Second*, we observe that fair ranking performance varies with respect to our six ground-truth population distributions. *DetConstSort* was able to achieve low NDKL scores for relatively-uniform distributions, like *A* and *B*, regardless of inference accuracy, but struggled to achieve high ABR scores at lower accuracies. Conversely, *DetConstSort* achieves relatively high ABR scores but low NDKL scores for three-group distributions that had an overwhelming majority group, like *C* and *D*. Distribution *E* appears to be a worst-case scenario, combining a clear majority group with three other, much smaller minority groups. These findings demonstrate that there are complex interactions between the composition of the underlying population, accuracy of inference, and fairness guarantees.

*Third*, by comparing the baseline NDKL and ABR values for non-fairness aware rankings in Table 1 to the fairness-aware results in Figure 4, we observe that there are cases where the former has better fairness scores than the latter, depending on the accuracy of demographic inference. This finding shows that the use of a fair ranking algorithm is not categorically better than

a non fairness-aware algorithm, depending on the accuracy of the underlying demographic data used for fair re-ranking.

*Fourth*, we observe no significant drop in the NDCG scores in Figure 4c for the fair ranked lists. This agrees with the findings of Geyik et al. [29] and demonstrates that utility need not be sacrificed to produce fair rankings. The decrease in NDCG scores is greater for population distributions like *D* and *F* that have greater skew, in contrast to more uniform distributions like *A* and *E*.

*Fifth*, we observe in Figure 4d that the MARC values of the list decreases as the inference accuracy increases. Lower MARC values signal smaller departures in ranking from the original list, highlighting again the pitfalls of imperfect inference. The decrease in MARC is less evident for skewed distributions.

### 5.2 Chess Ranking Dataset

We present the results of our first case study using the chess players dataset in Figure 5 and Figure 8a–8c. The former figure focuses on aggregate metrics, while the latter presents per-group metrics.

From the NDKL and ABR scores in Figure 5a and Figure 5b, respectively, we observe that the "fair" re-rankings that used inferred demographic data as input offer much worse fairness than the oracle; CNNG even has worse fairness than the unfair baseline. In contrast, the NDCG scores in Figure 5c demonstrate that the fair re-rankings have no impact on the utility of the results, irrespective of whether fairness is actually achieved.

The MARC scores in Figure 5d reveal that *DetConstSort* had to move candidates farther to achieve the fair (oracle) ranking for the chess case than in our other case studies (Figure 6d and Figure 7d). Further, we observe that *DetConstSort* moved candidates shorter distances when it was supplied with erroneous inferences, which helps explain why it failed to achieve oracle-level fairness.

To delve deeper, we look into the performance of the individual inference algorithms across the different sub-groups.[10] In the baseline unfair ranking, we observe that White and Asian males have high skew (Figure 8a), and that White males in particular

---

[10]The box plots follow standard statistical notation [75]. The box is bounded at the first and third quartile, and the central line represents the median. The upper whisker denotes the maximum point within the 3rd quartile + 1.5IQR (Inter Quartile Range), while the lower whisker denotes the minimum point within 1st quartile - 1.5IQR. The white dot denotes the mean.

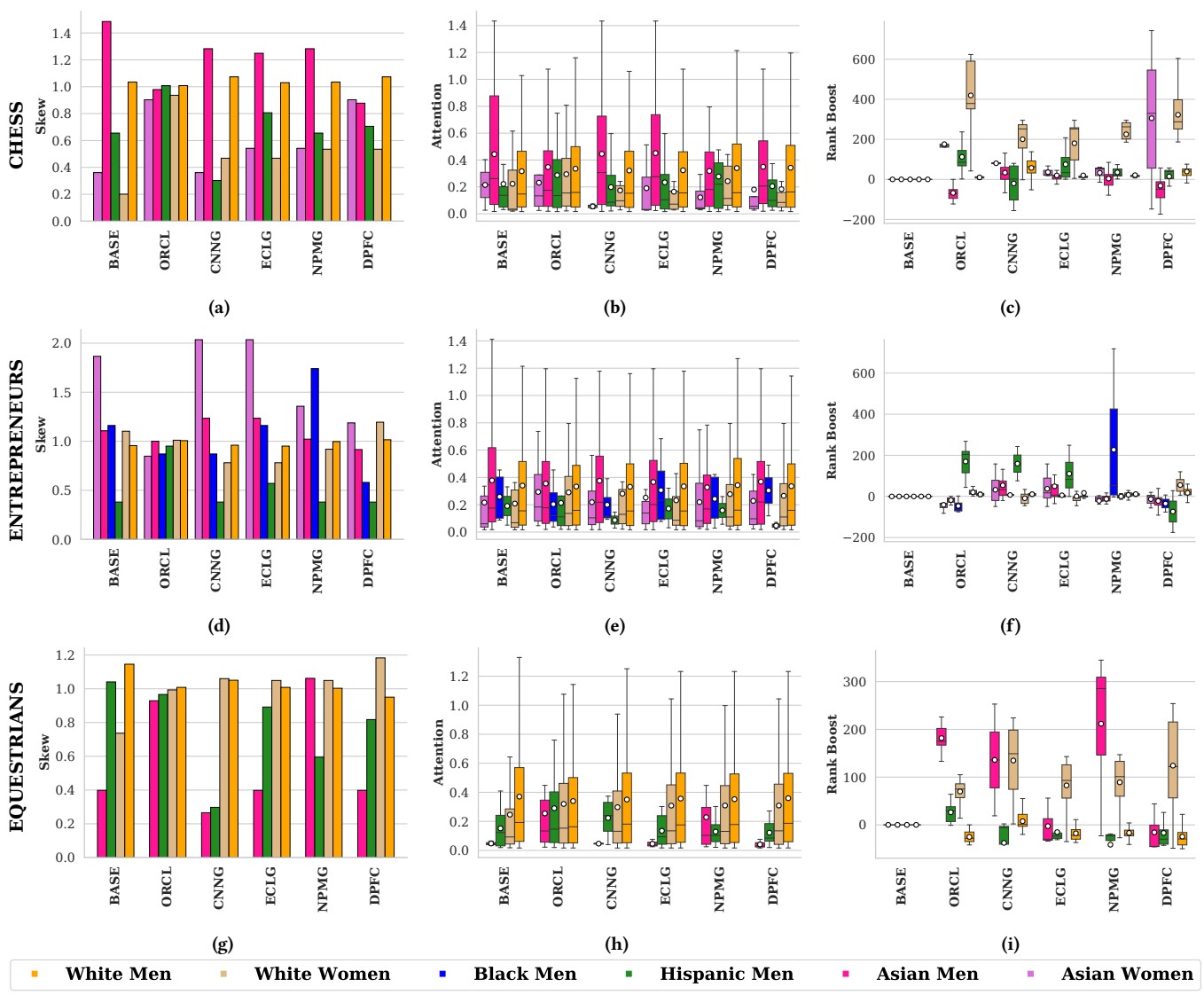

**Figure 8: Scores for individual groups in the Chess, Entrepreneurs and Equestrians datasets.**

receive a disproportionate amount of attention (Figure 8b). By design, when *DetConstSort* is given accurate demographic data it is able to produce a ranking with skew close to 1 for all groups, and attention is more uniform across the groups than in the baseline. Notably, *DetConstSort* had to dramatically increase the ranks of White women to achieve fairness (Figure 8c) because they are underrepresented in the population overall and especially within the top ranked players.

Conversely, we observe a variety of pathologies when *DetConstSort* is given inferred demographic data as input. Overall, we see that the advantaged groups (White and Asian men) retain their advantage, while non-White non-men rise or fall depending on the specific error characteristics of the inference algorithms. For example, the EthnCNN (CNNG) race inference algorithm mis-classified 15 White men and two Asian men as Hispanic men,

and mis-classified three Hispanic men as White men. These errors have the pernicious consequence of causing actual Hispanic men to be under-represented in the ranking—even below the baseline representation (Figure 8a).

Another example: Asian men appeared frequently at high ranks in the baseline ranking and thus *DetConstSort* attempts to decrease their representation. However, the name-based inference algorithms incorrectly label high-scoring Asian men, e.g., CNNG incorrectly labels Asian *men* as Asian (5) or White *women* (3). This increases the skew of Asian men at the expense of other groups (Figure 8a). Further, by mislabeling high-scoring men as women, this causes *DetConstSort* to provide lower rank boosts to women than in the oracle re-ranking (Figure 8c), and thus women receive lower attention than in the baseline and oracle rankings (Figure 8b).

## 5.3 Crunchbase Entrepreneurs Dataset

Of our three case studies, the entrepreneurs dataset is the fairest, i.e., the baseline and oracle NDKL and ABR scores are closest (Figure 6a and Figure 6b). Only Asian women and Hispanic men are over- and under-represented in the baseline ranking relative to the underlying population (Figure 8d). As expected, when given accurate demographic data, *DetConstSort* is able to equalize skew, although attention remains somewhat low for Black and Hispanic men relative to the other groups (Figure 8e).

We observe several notable artifacts in Figure 8d. Skew for Asian women increases when using EthCNN and EthniColor (Figure 8d) due to them being mislabeled as White women or Asian men. Further, being mislabeled into relatively larger groups causes the Asian women to receive less attention in the re-ranked lists (Figure 8e). Likewise, the low skew and low attention for Hispanic men can be attributed to the fact that all five of the people predicted to be Hispanic men by the inference algorithms were actually White or Asian men. Black men are over-represented for Nameprism (NPMG) because it mislabeled three high-scoring Black men as Asian/White men. To compensate, *DetConstSort* then moved two low-scoring Black men to very high ranks (as evinced by the large rank boosts in Figure 8f).

## 5.4 Equestrian Ranking Dataset

Our equestrian athlete case study contrasts our chess case study: in both instances we observe a large fairness disparity between the baseline and oracle (Figure 7a and Figure 7b), yet in the equestrian case the inference algorithms result in re-rankings that are closer to the oracle in terms of NDKL and ABR scores, whereas in chess the inference-driven re-rankings are closer to the baseline.

However, just because the inference-driven re-rankings are relatively fair on average does not mean individual groups are not being stigmatized. As shown in Figure 8g and 8h, Asian men have low skew and low attention. The oracle mitigates this issue, but the inference algorithms routinely mislabel Asian men and White men, thus resulting in Asian men having low skew and attention in the "fair" re-rankings as well. Nameprism is the exception: it predicted six out of seven Asian men correctly. Conversely, we observe cases where White women were mislabeled as Asian men, causing *DetConstSort* to dramatically increase their rank (Figure 8i), leading to cases where White women become over-represented.

## 6 DISCUSSION

In this study we investigate the interactions between five demographic inference algorithms and the *DetConstSort* fair ranking algorithm. To ensure realism, we derive the error rates for the demographic inference algorithms from real-world datasets, and present results from controlled simulations and real-world case studies. The takeaway from our experiments is that using inferred demographic data as input to fair ranking algorithms can invalidate their fairness guarantees in ways that are (1) difficult to predict and (2) often harm vulnerable groups of people.

It would have been a positive, pragmatic result if our study found that fair ranking under uncertain inference was categorically fairer than non-fairness-aware ranking, even if it did not achieve optimal fairness. Unfortunately, this is not the case: in some instances, groups that were not disadvantaged in the baseline, non-fairness aware ranking became disadvantaged under fair ranking due to errors in inference (e.g., Hispanic men in the Equestrians dataset).

## 6.1 Potential Mitigations

One tempting solution to the problem at hand is uncertainty-aware fair ranking algorithms [18]—if demographic uncertainty could be bounded, then the fair ranking algorithm could be modified to take this into account by adjusting the probabilities associated with each individual, thus producing fair rankings in expectation.

In practice, there are two shortcomings with this idea. *First*, the addition of uncertainty means that the ranking algorithm is unable to guarantee that any given realization of results is fair. Instead, the ranker only achieves fairness-on-average over the realization of many rankings [22]. Fairness-on-average may be acceptable in low-stakes situations (e.g., online dating) and unacceptable in high-stakes situations (e.g., resume search for hiring). *Second*, bounding the error rates for a demographic inference algorithm may not be feasible in practice. Even if the error rate can be measured for each protected group, the uncertainty-aware ranking algorithm must adopt the worst-case error rate from among the groups, i.e., convergence time for fairness-on-average is determined by the group with the greatest uncertainty.

The other solution is to intentionally collect demographic data, thus avoiding inference entirely. However, this data must be collected with great care and consideration. *First*, the choices presented to people (e.g., binary gender or US Census race/ethnicity categories) constrain the groups that can ultimately benefit from fairness interventions. *Second*, designers must determine whether self-reported or perceived demographic attributes are more appropriate for their context. For example, AirBNB purposefully uses perceived demographics to identify patterns of discrimination by hosts against guests [5]. *Third*, when classifying people we must always consider the potential for reifying oppressive structures [35]. In a given ranking context, if people are reluctant to divulge demographic data or there is the potential for this data to be misused, then designers must seriously consider whether algorithmically ranking people is appropriate in the first place.

## 6.2 Limitations & Future Work

The primary limitation of our work concerns how we operationalize gender, race, and ethnicity. Gender is not binary, but the sources of data we rely on (ground-truth and inferred) only support binary labels. Similarly, our work is constrained by the race and ethnicity categories that are supported by available inference algorithms. These categories lack nuance and reify problematic political hierarchies. Future work in this space should broaden the space of gender, racial, and ethnic categories that are critically examined [32, 37], as well as examine other marginalized communities.

## ACKNOWLEDGMENTS

The authors thank Carolyn Rose and the anonymous reviewers for their comments. This work was supported by a 2019 Sloan Fellowship award, NSF grants IIS 1917668 and IIS 1822831, and Dow Chemical. Any opinions, findings, and conclusions or recommendations expressed in this material are those of the authors and do not necessarily reflect the views of the funders.

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
