# OpenReview forum: "When Fair Ranking Meets Uncertain Inference"
_ACM.org/SIGIR/Badging_

### Official Review · ~Anna_Ruggero1 · 2021-07-08
**Approved**

**Comment:**

Hi authors,

I had a look at the latest changes:
- Requirements file: OK
- Addition of the file ../chess_nameprism_pred.p: OK
- y-axis: OK
- documentation: OK (minimal)
- GIT tag: OK

You fulfill all the requirements for the two Functional and Reusable/Available badges but I still suggest spending a few more words on the documentation.

Congratulations!

**Awarded Badges:**

["Artifacts Evaluated – Functional", "Artifacts Evaluated – Reusable and Available"]

---

### Official Review · ~Charles_Clarke1 · 2021-07-08
**Approved**

**Comment:**

I worked with Anna Ruggero, and I also independently downloaded and inspected the data and ran the code.

With the corrections to the minor issues, I am happy to award the badges.

Congratulations!

**Awarded Badges:**

["Artifacts Evaluated – Functional", "Artifacts Evaluated – Reusable and Available"]

---

### Official Review · Program_Chairs · 2021-07-09
**Functional, Reusable, and Available Badges**

**Comment:**

Dear Authors,

According to the discussion you had with the reviewers and to the final reviews, your artifact is ready for badging.

We are happy to award you the following badges:
* Artifacts Evaluated – Functional
* Artifacts Evaluated – Reusable and Available

**Awarded Badges:**

["Artifacts Evaluated – Functional", "Artifacts Evaluated – Reusable and Available"]

---

### Public Comment · ~Anna_Ruggero1 · 2021-06-29
**Necessary changes**

Hi authors,

These are few things missing in order to achieve the first 2 badges (Artifacts Evaluated – Functional and Artifacts Evaluated – Reusable and Available):
- A requirements file is missing
- The file ../chess_nameprism_pred.p necessary to cell 39 is missing (consequently not all the plots are available)
- The y-axis of the plots is too much approximate, therefore we have not a plot identical to the one of the paper (there are cases where we read only zeros or ones in the y-axis).
- We think that the code itself lacks documentation. There is no explanation about the structure of the main jupyter file, even if it is subdivided through titles. There are very few comments in the code and there is no explanation of what a function should do (we have deduced it just from the functions' name).
- The code version is not tagged in GIT

When you will address these issues (which shouldn't take long) we will award the badges.

---

> ### Public Comment · ~Avijit_Ghosh1 · 2021-06-29
> **Made changes**
>
> Thank you for the feedback. I have made the necessary changes and added comments. I also created a tag for the code on github: https://github.com/evijit/SIGIR_FairRanking_UncertainInference/releases/tag/v1.1
>
> Please let me know if you need anything else for the badging. Thanks.